# Reproductive Biology and Pollination Ecology of *Fritillaria michailovskyi* Fomin (Liliaceae), Endemic to East Anatolia (Turkey)

Meral Aslay [1], Faruk Yıldız [2] , Ozkan Kaya [1,*] and Claudia Bita-Nicolae [3,*]

1   Erzincan Horticultural Research Institute, Republic of Turkey Ministry of Agriculture and Forestry, 24060 Erzincan, Turkey
2   Institute of Science, Erzincan Binali Yıldırım University, 24002 Erzincan, Turkey
3   Ecology, Taxonomy & Nature Conservation Department, Institute of Biology Bucharest, Romanian Academy, 060031 Bucharest, Romania
*   Correspondence: kayaozkan25@hotmail.com (O.K.); claudia.bita@ibiol.ro (C.B.-N.)

**Abstract:** *Fritillaria* is highly endangered in their natural habitats, and these species are perennial bulbous plants with an important medicinal and ornamental value whose reproductive strategies and adaptive evolution mechanisms are still not fully clear. Therefore, the reproductive strategies of endemic species, like *Fritillaria michailovskyi* Fomin are important to detect the community structure and the diversity patterns of ornamental plants. The current paper on the reproductive strategy of *F. michailovskyi*, a rare endemic species, was carried out at the Erzincan Horticultural Research Institute, Turkey. Our results indicate that the flowering stages of *F. michailovskyi* may be divided into eight phases. According to pollination experiments and the pollen/ovule ratio, and the self-incompatibility index (SII) in an ex-situ population, *F. michailovskyi* indicated high levels of xenogamy and self-incompatibility. It was determined that the pollination of *F. michailovskyi* mostly depended on pollen vectors, and the effective pollinators of *F. michailovskyi* were *Apis mellifera* and *Bombus terrestris*. In addition, average seed number, seed germination, and average seed viability were found as 144, 46%, and 67%, respectively. The stigma receptivity, pollen grains, and pollen viability were detected as 83%, 252,000, and 95%, respectively. Our study is the first report providing a detailed explanation of the reproductive strategy of this rare endemic species, which could aid in the genetic evolution and conservation of this valuable taxa.

**Keywords:** flowering biology; endemic species; pollination biology; honeybee; pollinator

## 1. Introduction

Reproduction in plants is the life cycle that is necessary for the perpetuation of the species, and genetic variation is primarily produced via recombination steps in the sexual breeding of plants, which is, so, a process of a key role in terms of species and population biology [1,2]. Reproduction, in other words, is not only the core of plants' evolutionary process but also a comparatively fragile process in their life [3]. The comprehensive information on their reproductive strategy is, therefore, required for the successful cultivation and preservation of plants, and the character studies on the reproductive strategy of this species are important in terms of discovering the mechanisms by which its extinction is endangered [3,4]. The reproductive biology of plants primarily focuses on floral biology, gene flow, flowering phenology, pollen-pollinator interaction, and breeding systems through seeds and pollen [5]. In this stage, observing and recording the periodic growth stages in the phenological events of plants and examining the dependence and regularity of annual development cycles on environmental conditions is the basic step. Considering phenological events in plants (i.e., leaf expansion, budburst, flowering, leaf-abscission, seed set, fertilization, fruiting, seed germination, and seed dispersal), they all follow each other

in a sequential manner in a season. Numerous studies have been, indeed, reported on plant breeding systems and phenological events ranging from full self-incompatibility to full self-compatibility [6–8]. Self-compatibility for the species removes mate restraint by allowing each plant flower to pollinate itself, whereas self-incompatibility within some plant species decreases the problem of inbreeding depression by restricting the number of compatible mating pairs [2]. There is the phenomenon that open-pollinated flowers of hermaphrodite plants, which often observe self-incompatibility, will have a very low fruit set [9]. In general, a low fruit set may be mostly because of a high level of self-incompatibility and a high incidence of self-pollination in nature; however, several different reasons, namely resource limitation and location of fruit within inflorescences, can be involved as well [10]. Moreover, the study of plant breeding systems can also provide valuable information on the evolution of plant mating systems and the potential for plants to adapt to changing environmental conditions through shifts in breeding systems.

Although about 600 species are known to have disappeared in the last 250 years, it is currently estimated that approximately 20 to 39% of plant diversity is at risk of extinction [11]. Regarding plants, only 10% of these species have been reported to be at risk of extinction globally and have been defined by the International Union for Conservation of Nature (IUCN), Red List [12]. A widely distributed member of the Liliaceae, which is the genus *Fritillaria* L., consists of about 160 species occurring in many parts of the northern hemisphere, with centers of speciation in Greece and Turkey, Western North, Iran, America, and East Asia [1,13–15]. *Fritillaria* is found in different habitats and is a cultivar of climatic areas, and it is located in a wide latitude range from riparian regions, deserts, coasts, steppe, woodland, meadows, alpine zones, and mountain screes [16]. Some *Fritillaria* L. is a seriously endangered species, according to IUCN Red List Categories and Criteria [17]. Demirkuş et al. [18] have also reported that the IUCN category of *Fritillaria michailovskyi* Fomin is Conservation Dependent (CD). Natural regeneration of the species is relatively poor, and seedlings are few in each population. It should be emphasized that as the remaining partial populations are severely damaged by human activities, their status is a cause for serious concern, and these wild resources need urgent protection. These species are bulbiferous, bloom in the spring after the snow melts, and produces an erect flowering stem, either multi-flowered racemes or a single flower [19]. The flower of species is usually actinomorphic and has a typical campanulate perianth, trimerous, and tulip-like, but with flowers facing the ground after flowering [20,21]. In Turkey, which has an important place in terms of the distribution of this genus, 47 species belonging to the genus *Fritillaria* show natural habitat [22–24].

*F. michailovskyi*, which is included within this genus, is an endemic species belonging to Turkey and has distribution in the provinces of Ağrı, Van, Kars, and Erzurum [21,25]. This species is easy to grow, and it is a highly exotic species thanks to the attractiveness of its flowers. Although many *Fritillaria* species reach flowering in 5–6 years from seed sowing, *F. michailovskyi* species can bloom in 3 years. Its flowers are purple, and the tips are yellow-striped, as well as flower shape bell-shaped [26,27]. Given that these characteristics, flower size, and color, as well as being involved in protection and attraction, could also signal to inform potential floral visitors about the characteristics of the present reward. It is considered that bumblebee pollination is the most basic vector, though no pollinator record was recorded for *F. michailovskyi* in the nature conations. This assumption can be confirmed with the presence of a landing platform for *F. michailovskyi* with large pendulous flowers. Although the data is limited, flower visitors are attracted by a combination of nectars and visuals, and one might assume this is true for *F. michailovskyi.* However, the relationship between nectars and visual pollination strategies and its results on the breeding systems within *F. michailovskyi* is still obscure, and no information is available. This study represents the first paper on the reproductive strategies of *F. michailovskyi*, and hence the objectives of our research were (1) to define seed and pollen characteristics; (2) to determine reproductive strategies; (3) to detect flowering phenology; (4) to verify insect visitor identities.

## 2. Materials and Methods

### 2.1. Plant Material and Study Site

Data on the reproductive strategy of *F. michailovskyi* were gathered during the 2019, 2020, and 2021 flowering seasons. Plant material utilized for reproductive biology was obtained from *F. michailovskyi* cultivated in ex-situ at Erzincan Horticultural Research Institute, Erzincan, Turkiye (39°75′ E, 39°36′ N. 1307 m asl), from the private collections of the co-author Meral Aslay. Some flower characteristics and odor, nectar, reproductive organs parameters (size of the leaf size, flowering, and vegetation time; number, color, size, and odor of tepals; color, number, plant height, and width) of *F. michailovskyi* were previously reported by Aslay et al. [27]. The flower number of *F. michailovskyi*, typically in ex-situ, varies from 1 to 13 per plant. The tepal ends of its flowers are straight in some populations and curved backward in some populations, with the claret stripe on yellow sepals and multi-flower and perianth segments yellow at the apex and brownish purple at the base. In the phenological phases of species, how this phase corresponds to the anthers and how long it lasts and pistil fertility stages, and when the flowers begin to secrete nectar, where nectars are found was performed by considering the reports of previous researchers [20,21]. Depending on the climatic conditions, the flowering time continues from the last week of March to the middle of April, and this period lasts for 16–20 days. Plant height is almost 10–32 cm, and the mean values of bulb diameter, leaf width, and leaf length of *F. michailovskyi* were measured as 25, 12, and 62 mm, respectively. The inner tepal length and inner tepal width, outer tepal width, and outer tepal length of *F. michailovskyi* were measured as 11, 27, 10, and 24 mm, respectively [27].

### 2.2. Detection of Pollination Strategies

This study randomly marked 290 individuals at the tepal unopened phase to detect the pollination strategy of the examined population, and these are followed by five applications; (i) open-pollination (flowers were not manipulated), (n = 50), (ii) spontaneous self-pollination, in which flowers were bagged with mesh bags (n = 50), (iii) spontaneous cross-pollination, in which flowers were emasculated; (n = 50), (iv) induced self-pollination in which flowers were emasculated bagged with mash bags and artificially self-pollinated 3–5 days after anthesis before being re-bagged (n = 50), (v) xenogamous pollination by hand (and bagged with mesh bags, flower buds were emasculated and, flowers of the plant were pollinated utilizing pollen from different plants at least 5–7 m apart 4–6 days after anthesis before being re-bagged (n = 50), (vi) apomixis: induced cross-pollination, in which flowers were emasculated and bagged with mesh bags (n = 40). For hand pollination based on Wang et al., [28] and Dafni [29], we utilized either another flower from a plant at a distance of nearly 3–4 m (cross-pollination) or self from one dehisced anther of the same plant flower (self-pollination). To guarantee the reproductive strategy in induced self-pollination, the pollination experiments were repeated daily during the flowering period (10 days) to avoid missing the periods when pollen and stigma receptivity were at their peak. The flowers utilized in the experiment were marked with tags, and the fruit set was detected in May when the seeds were mature. The fruit set of flowers = the number of mature fruits/the number of flowers treated × 100%.

### 2.3. Detection of Flowering Dynamics

Considering our observations, the flowering stage of the *F. michailovskyi* population was roundly 3 weeks. Since flowering dynamics will be reported for the first time in this study, its stages were classified into eight stages; (i) bud burst; (ii) flowering initial; (iii) pre-dehiscence phase; (iv) first dehiscence phase; (v) full dehiscence; (vi) end-dehiscence phase; (vii) fresh perianth phase and (viii) perianth wilting phase (Figure 1), based on the methodology used by Yildiz et al. [1] for *F. aurea*.

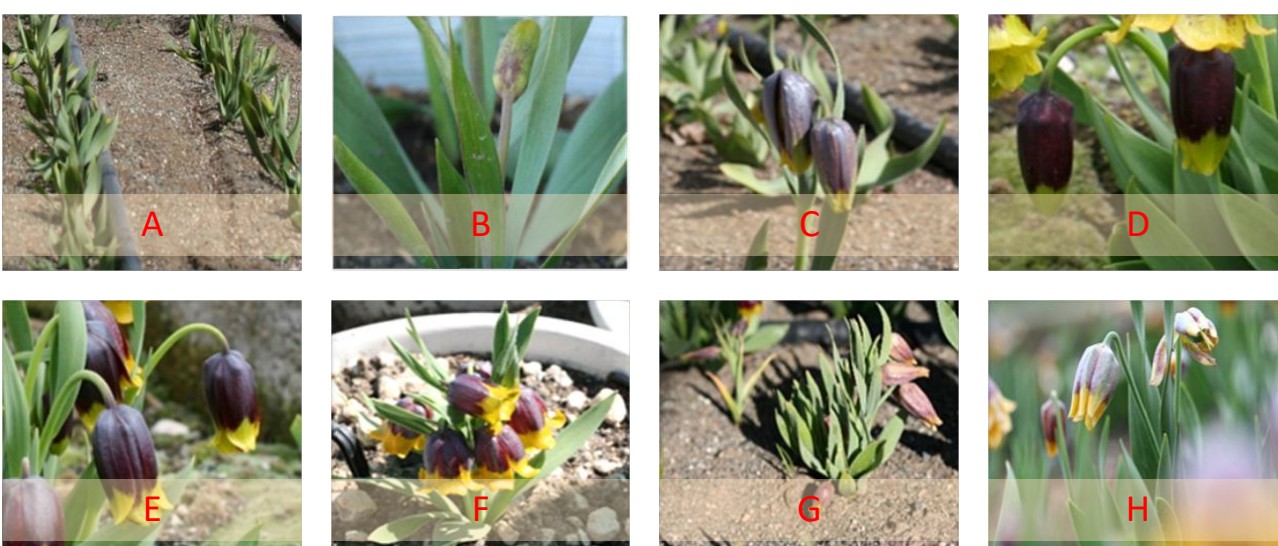

**Figure 1.** Flowering dynamics; (**A**). bud burst, (**B**). flowering initial, (**C**). pre-dehiscence phase, (**D**). first dehiscence phase, (**E**). full dehiscence, (**F**). end-dehiscence phase, (**G**). fresh perianth phase and (**H**). perianth wilting phase. (original photo by M. Aslay).

### 2.4. Detection of Pollen Viability and Stigma Receptivity

We randomly marked five flowers of *F. michailovskyi* at each of the six phenological phases. Pollen viability and stigma receptivity at each developmental stage were evaluated by considering the morphological changes in male and female organs within the 2-week life cycle of the plant flower (i.e., the fresh perianth, end-dehiscence stage, full dehiscence, first dehiscence stage, pre-dehiscence stage, and stage initial flowering). Stigma receptivity of *F. michailovskyi* was detected utilizing the $C_{12}H_{12}N_2$-$H_2O_2$ (benzidine-hydrogen peroxide reaction solution) based on Dafni and Maués [30] method. For stigma receptivity, the $C_{12}H_{12}N_2$-$H_2O_2$ reaction solution was conducted at a ratio of water:3% $H_2O_2$:1% $C_{12}H_{12}N_2$ (22:11:4). Marked *F. michailovskyi* stigmas were placed on a petri dish, an appropriate amount of the $C_{12}H_{12}N_2$-$H_2O_2$ was dripped onto samples, and then the amount of bubble formation on the stigma was observed. For the stigma receptivity, discoloration and bubbling around it were observed under the microscope, and images were recorded after 1 min. The receptivity of the stigma was estimated according to bubble densities around the stigma observed under the microscope based on Dafni and Maués [30] procedure, as follows: (i) stigma receptivity weak (bubble formation at roughly 5–10% of the stigma surface or if bubble density is very low), (ii) receptivity middle (bubble formation at roughly 45–50% of the stigma surface or if bubbles are moderate), (iii) stigma receptivity strong (bubble formation at roughly 85–90% of the stigma surface or if the bubbles are dense).

Ten flowers of *F. michailovskyi* were randomly used at the pre-dehiscence phase to determine pollen viability based on the acetocarmine staining method [31]. The 1% aceto-carmine was added to the anther sacs left on the slide to disperse the pollen, and then the anther sacs were removed from the slide. The fresh pollen sacs immediately obtained from the flowers were spread on acetocarmine and then covered with a coverslip and heated at 50 °C for 1 min to allow the dye to diffuse through the pollen grains. Afterward, the color change was observed by a 10× zoom under the light microscope in the pollen grains. Undyed grains were accepted as not viable, light red grains were accepted as semi-viable, and dark red grains were accepted as viable [1,31].

### 2.5. Detection of the Self-Incompatibility Index (SII) and Pollen/Ovule Ratio (P/O)

Ten different individuals of the *F. michailovskyi* plant in ex-situ were randomly selected, and one anther was removed from each of these individuals at the pre-dehiscence. Pollen in the anthers was then spread on a slide with a counting scale (10 × 10 mm), and the pollen grains of the samples were counted under a stereomicroscope. Ovules were randomly

selected from ten different plants at the beginning of flowering to determine the number of ovules, and then young ovules, which were divided into two with a scalpel on the slide, were removed in groups. Afterward, these ovules were examined under a stereomicroscope and counted. The number of pollen grains produced for each ovule in a flower was determined using Cruden's [32] method. For the SII of the *F. michailovskyi* flower, seed setting success of self-pollinated and cross-pollinated plants was recorded after flowering, and regarding SII, the self-incompatibility index was calculated based on Zapata and Arroyo [33] method.

### 2.6. Detection of Seed Viability and Germination

After about 28–30 days from the perianth wilting phase, we randomly marked ten fruits of *F. michailovskyi*, and seed viabilities of plants were performed by the 2.3.5-triphenyl tetrazolium chloride test (TTC) according to Dafni [29]. Three hundred seeds obtained from fruits were exposed to the TTC solution. Seeds in capsules were then submerged in distilled water to activate the embryo for 24 h, and these seeds were exposed to a 0.1% solution at 22–24 °C for 24 h. Embryos in halved seeds were examined after adequate incubation time. For the seeds examined under the stereomicroscope, embryos without any color change were accepted as dead seeds, and those with red color were accepted as viable seeds. Besides, seed germination trials were conducted according to Aslay et al. [34].

### 2.7. Detection of Floral Visitors

For the studies, pollinator behavior was determined by photographing insects visiting flowers during full bloom for twenty consecutive days (from the last week of March to the middle of April) without rain (on sunny days) in 2019–2021. Insect visits were recorded every hour from 8:00 am to 16:00 pm for each observation day (based on some preliminary study observations carried out at sunset and in the early morning, they were not recorded as they showed no insect activity). Each type of recording (lasting approximately 1 h) followed three stages: (i) video recording based on insect activity (with Canon 5D Mark IV camera in a 30 min digital video), (ii) the random selection of flowers according to insect visitation and (iii) insect capture (Supplementary Materials). Different flower pollinators visiting flowers for insect identification were collected utilizing a trap (atrap), and a sample was taken from each to be identified. Effective floral visitors were detected as insects that not only deposited it on a receptive stigma but also picked up pollen based on the evaluation criteria of Stout [7]. Therefore, the effective floral visitors for *F. michailovskyi* were defined, and their visiting behaviors were pictured and recorded.

## 3. Results

### 3.1. Floral Features and Reproductive Phenology of F. michailovskyi

Some floral features of *F. michailovskyi* reported for the first time in this study are shown in Table 1. The inflorescence is broadly campanulate (Figure 1B), and the total number of flowers per inflorescence ranged from 1–13. The inflorescence has changed between 1–13 branches with similar characteristics, and the flower in the plant was arranged in a gamosepalous succession on it. The flower of the plant was hermaphrodite and actinomorphic, odorless, yellow at apex, brownish purple at base, in color with an average length of 16–33 mm. The nectar sacs had a greenish, and anther dehiscence consisted of bursting inwards by slits with an average length of 5–11 mm. Corolla consisted of six yellow at the apex and brownish purple at the base, with an average length of 18–36 mm. In the flower, the androecium consisted of six stamens located in a ring around the carpel, and they were present at the above level of stigma (Figure 2c). The average number of pollen grains per flower, the number of ovary capsule lobes, the average number of ovules per ovary, the average length of filament, the mean length of anther average length of stamen, and the average length of style were detected as 252,000, 6, 144, 8 mm, 8 mm, 11 mm, and 8 mm, respectively. The filament was short and greenish-creamy in color, while anthers were yellowish and basifixed. The anther was dehisced between 09:30 am–13:30 pm, and

this dehiscence occurred through a slit in the vertical plane (Figure 2c). In general, the anthers dehisced nearly 40–45 min after flower anthesis and shed 70% of the pollen within approximately 3–4 h. The non-fruiting flowers are abscised in 6–7 days. Then the tepals wilted and fell off, and the stamens dried from the basal part of the ovary (Figure 1H). The ovary of the plant remained attached to the pedicel after pollination, but parts of the style and stigma dried up. Pollen surface, pollen shape, pollen cell, pollen size, and pollen viability were determined as granulate, sub prolate 2 celled, 31–66 μm, and 95%, respectively (Table 1). Stigma shape, stigma surface, the average diameter of the ovary, ovary placentation type, ovary status, ovary capsule, stamen type, and perianth type were detected as trifid or trilobate, wet, 2.1 mm, axile, 6, hypogin, sin genesis stamen, and perigon, respectively.

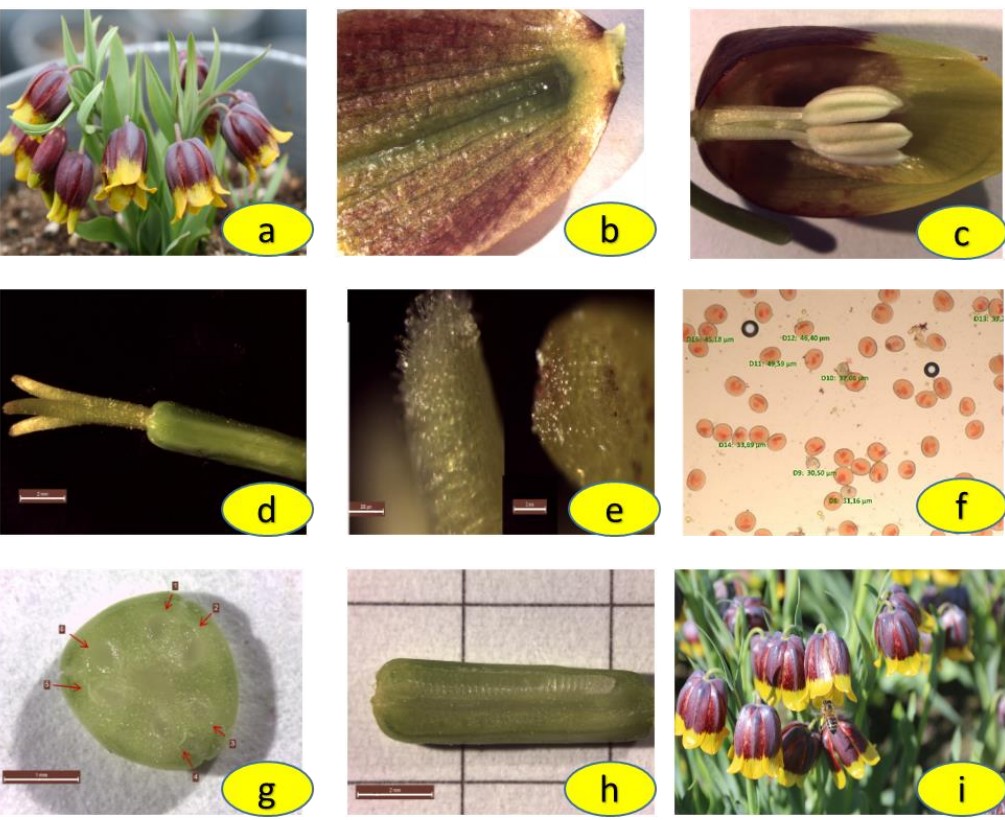

**Figure 2.** Floral features and reproductive phenology of *F. michailovskyi*. (**a**). F. michailovskyi, (**b**). nectary sac, (**c**). anthers, (**d**). pistil, (**e**). stigma receptivity, (**f**). pollen grains, (**g**). seed capsule (arrows represent the location of the six seed capsules), (**h**). seed set, (**i**). pollinator (*Apis mellifera*). (original photo by M. Aslay).

**Table 1.** Some identified floral features of *F. michailovskyi*.

| Parameters Studied | Observations |
|---|---|
| Flowering period | March–April |
| Inflorescence | Broadly campanulate |
| Flower | Hermaphrodite and actinomorphic |
| Flower color | Purple-stricted yellow |
| Flower length | 16–33 mm |
| Odor | Nil |
| Nectar | Greenish, Oblanceolate |
| Anther dehiscence | Bursting inwards by slits |
| Number of anthers/flower | 6 |
| The average number of pollen grains/flower | 252,000 |
| The average number of ovules/ovary | 144 |

**Table 1.** *Cont.*

| Parameters Studied | Observations |
| --- | --- |
| Length of tepal | 18–36 mm |
| Length of filament | 6–10 mm |
| Length of anther | 5–11 mm |
| Length of stamen | 8–14 mm |
| Length of style | 6–10 mm |
| Pollen-ovule ratio | 1750 |
| Pollen surface | Granulate |
| Pollen shape | Subprolate |
| Pollen cell | 2 celled |
| Pollen size | 31–66 μm |
| Pollen viability | 95% |
| Stigma shape | Trifid or trilobate |
| Stigma surface | Wet |
| Diameter of ovary | 1.8–2.3 mm |
| Ovary placentation type | Axile |
| Ovary status | Hypogin |
| Ovary capsule | 6-lobed |
| Stamen type | Singenesis stamen |
| Periant type | Perigon (Tepal) |
| Corolla type | Gamosepal |
| Anther type | Basifix |
| Perianth estivation | Imbricate-alternate |
| Anther dehiscence direction | Extrorse |

### 3.2. Flowering Phenology of F. michailovskyi

From the beginning of March to the end of March, the first flowers on the dormant buds of *F. michailovskyi* are formed underground and are emerged together with the flowers from late March to mid-April (Figure 1A). The flowers of the plant opened approximately in 9 days, and the pollen presentation of the flowers continued for an average of 5–7 days. The flowering peak was determined from the second week of April to the end of April. The flowering process of the plant continued for nearly 18–22 days, as follows: (i) the bud phase on soil, (ii) the initial flowering time (6–9 days), (iii) pre-dehiscence tepal coloration (3–4 days), (iv) full dehiscence (3–5 days), (v) end-dehiscence (4–5 days), (vi) the fresh tepal of flowers (6–8 days), (vii) the tepal wilting time (4–5 days), and (viii) after the dried tepal, the emergence of seed capsule (10–20 days) (Figure 1A–H). Flowering showed a decrease from the last week of March, and it was observed that the flowers disappeared completely in the end of April.

### 3.3. Self-Incompatibility Index (SII) and Pollen/Ovule (P/O) Ratio of F. michailovskyi

Considering the pollination strategies of *F. michailovskyi*, the SII ratio could not be determined as zero (0) since self-pollination did not occur. On the other hand, P/O has been widely utilized in traditional breeding studies as a rough estimator of reproductive system studies. Plant reproductive systems have been reported to be associated with certain flower traits. The average ovule and pollen numbers per flower of *F. michailovskyi* were utilized to calculate the P/O ratio in 2019–2021. The number of ovules per flower varied between 108 and 234 (average 144). Total pollen production per flower averaged 252,000. In the current study, the P/O ratio of *F. michailovskyi* was 1750 (Table 1).

### 3.4. Pollen Viability and Stigma Receptivity of F. michailovskyi

Pollen viability of *F. michailovskyi* was detected at 95% during anther dissociation using acetocarmine staining (Table 1 and Figure 2f). Considering the flowering stages, the pollen viability ratio ranged between 63 to 95% according to the stages. Pollen viability showed a steady increase during flower development stages in the ex-situ conditions, and viability reached less than after the tepal wilting stage (Figure 3). On the other hand, stigma

receptivity was determined as 0, 15, 35, 53, 90, 95, 72, 55, and 25% at bud burst, flowering initial, pre-dehiscence stage, first dehiscence stage, full dehiscence, end-dehiscence stage, fresh tepal stage, and tepal wilting stage, respectively. The data of the stigma receptivity test indicated that the stigma was not receptive on the first day of anthesis or during the sympetalous period, as they showed no staining reaction. Stigma secretion of flowers increased. Then reacted more with the $C_{12}H_{12}N_2$-$H_2O_2$ reaction solution 2–6 days after the anthesis, i.e., blue stigma staining and bubbles became more prominent. At 5–7 days, receptivity was highest, during which time the secretions of the papilla were most abundant, and after 6 days, receptivity declined rapidly (Figure 3).

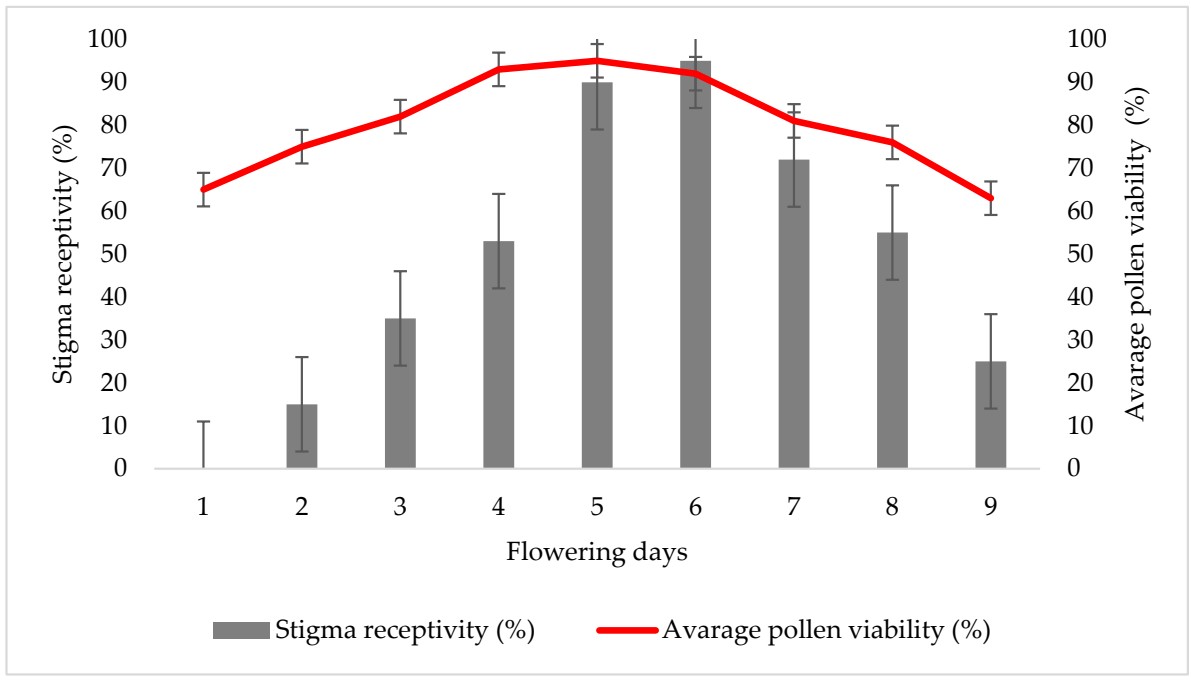

**Figure 3.** Changes of pollen viability and stigma receptivity during anthesis of *F. michailovskyi*.

### 3.5. Pollination Experiments of F. michailovskyi

In current work examined six pollination types on *F. michailovskyi*. It was observed that the fruit set rate varied between them: xenogamous pollination (86%) > spontaneous cross-pollination (82%) > open pollination (66%) > spontaneous self-pollination (0) = induced self-pollination (0) = apomixis (0) (Table 2). It was observed that none of the bagged and emasculated flowers, as well as those flowers bagged without castration, set fruit. The seed length of the plants ranged from ~3.0 to 5.0 mm. Based on our findings, the fruit setting rate of open-pollination and spontaneous cross-pollination were only slightly lower (6 and 20%) than that of xenogamous pollination, respectively (Table 2). In addition, the average number of mature seeds in each flower was 59, 60, and 68% in xenogamous pollination, open-pollination spontaneous, and cross-pollination, respectively. The ovoid capsule of the plant flower was dehisced by six equal valves (Figure 2g). The number of seeds per capsule varied from 18–39 per capsule. However, most of the ovary per flower contained an average of 144 seeds nearly (Figure 2h).

### 3.6. Germination and Seed Viability of F. michailovskyi

The seed viability of the flower was observed only in spontaneous cross-pollination, xenogamous pollination, and open-pollination applications because the fruit set was not determined in other pollination treatments. Regarding the findings of the TTC test, viable ratios of seeds in xenogamous pollination, open-pollination spontaneous, and cross-pollination were 74, 63, and 64%, respectively (Table 2). In our results, we observed

differences in seed germination between spontaneous cross-pollination, open-pollination, xenogamous pollination, and with rates of 45, 43, and 51%, respectively.

**Table 2.** Fruit set of Fritillaria michailovskyi under different pollination experiments.

| Treatment | Flowers | Fruits Developed (%) | Fruit Set (%) | Seed Viability (%) | Germination (%) |
|---|---|---|---|---|---|
| Open-pollination | 50 | 33 (66) | 68 | 63 | 43 |
| Spontaneous self-pollination | 50 | 0 | - | - | - |
| Spontaneous cross-pollination | 50 | 41 (82) | 60 | 64 | 45 |
| Induced self-pollination | 50 | 0 | - | - | - |
| Xenogamous pollination | 50 | 43 (86) | 59 | 74 | 51 |
| Apomixis | 30 | 0 | - | - | - |

*3.7. Pollinator Behavior Observation of F. michailovskyi*

The flowers *of F. michailovskyi* were visited by *Apis mellifera*, *Bombus terrestris* (Hymenoptera), and moths (Lepidoptera). Given our observations, it may be realized that *A. mellifera* and *B. terrestris* visited the flower in the morning (9:30 am and 12:00 pm) and afternoon (13:00 pm and 15:00 pm) after anthesis. *A. mellifera* and *B. terrestris* (Hymenoptera) played a key role in xenogamous pollination, open pollination, spontaneous, and cross-pollination. The average number of visitors for the most intense flowering period in the 20 different plants examined is presented in Figure 4. Since the moths were observed only twice a day and for a very short time, they were not considered effective pollinators.

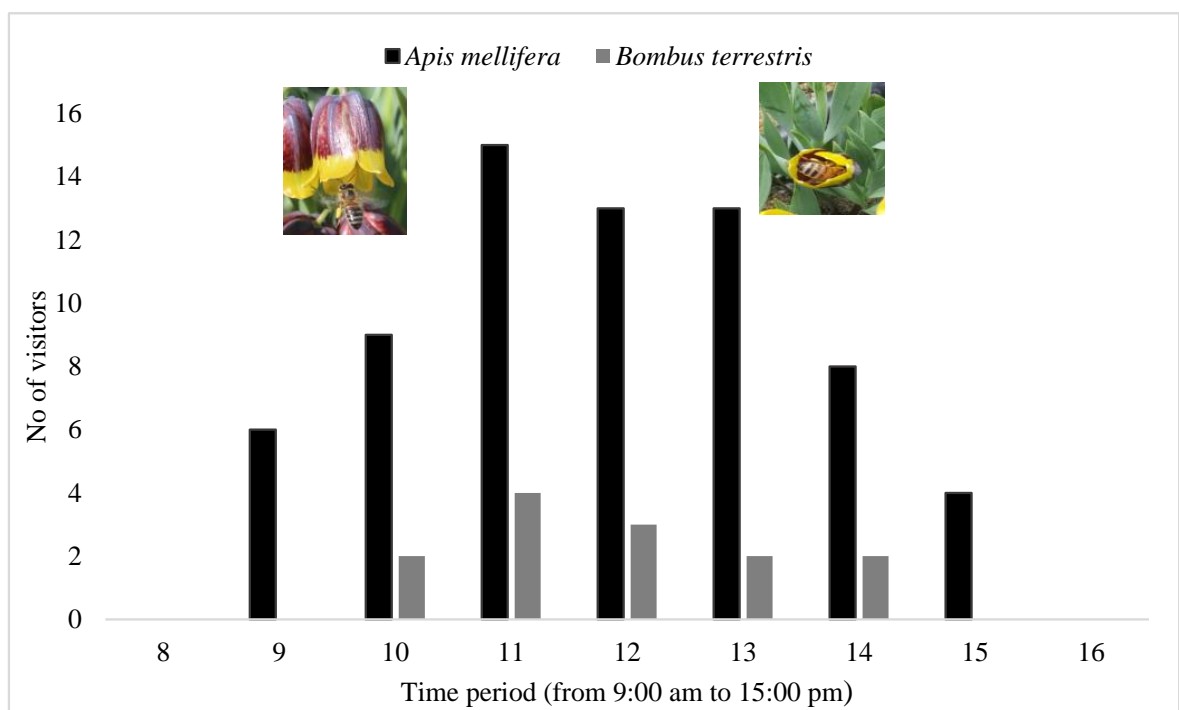

**Figure 4.** The average number and visiting periods of floral visitors.

## 4. Discussion

Though partial development has been revealed in understanding the strategies of several *Fritillaria* species on reproductive biology and flower traits [1,5,27,35], a question remains unclear among some researchers over the past few decades: What are the factors affecting the reproductive strategies and flower traits of threatened and endemic plants with very few populations, such as *F. michailovskyi*? Indeed, despite some studies on the flower characteristics of some *Fritillaria* species [1,27,34,36], the terminology of the flower morphology of *F. michailovskyi* has not been sufficiently clarified.

The morphological features of reproductive organs in *F. michailovskyi* have been, therefore, extensively reported with concepts together in plant terminology (Table 1), different from previous studies. In our study, flower characteristics such as anther dehiscence, number of anthers/flower, stigma shape, pistil type, ovary placentation type, ovary status, ovary capsule, stamen type, periant type, anther type, anther dehiscence were similar to many *Fritillaria* species [1,20,36–38]. Besides, flower characteristics (flowering period, inflorescence, flower, length of tepal, length of filament, flower color, flower length) were different to many *Fritillaria* species [1,20,37–42]. Odor, nectar, reproductive organs parameters (average number of ovules/ovary, the mean number of pollen grains per flower, length of anther, length of stamen, length of style, pollen-ovule ratio, pollen surface, pollen shape, pollen cell, pollen size, pollen viability, stigma surface) were different to many *Fritillaria* species [1,20,37–42]. As far as our knowledge, on the other hand, this is the first record of flower characteristics, odor, nectar, and reproductive organs in *F. michailovskyi* (Table 1), and thus, it was very difficult to compare these results with the literature.

It is crucial to take action to conserve plant diversity as they play a crucial role in maintaining the balance of ecosystems and providing vital ecosystem services. General knowledge of reproductive strategy in plants plays a key role not only in systematic studies and evolution [43] but also ineffective conservation strategies [44] for threatened and endemic plants with very few populations, such as *F. michailovskyi*. Considering reproduction between the most important phases in the life cycle of plant species, to date, there is no data on the dispersal of seeds and germination followed by the seedling establishment to members of *F. michailovskyi*. Information on reproductive stages is an important step when it comes to resource allocation in a plant's life cycle, as assimilates previously reserved for vegetative growth in the plant will be used for generative purposes [45]. Although this is the first comprehensive work on reproductive strategy in *F. michailovskyi*, flowers were generally nodding and bell-shaped, which has been stated in other *Fritillaria* species [35]. In consequence, anthers and nectarines were not visible to approaching pollinators. However, *F. michailovskyi* has higher parental traits compared to many other *Fritillaria* species (unpublished data). It can thus be considered an important species in *Fritillaria* breeding, both as a female parent and as a male parent. In fact, fruit set results on xenogamous pollination (86%), spontaneous cross-pollination (82%), and open pollination (66%) applications support these findings (Table 2). These results suggest that weak pollen limitation occurred in *F. michailovskyi* populations, which is contrary to the effective pollen limitation found in some *Fritillaria* species, namely *F. delavayi* Franch, *F. maximowiczii* Freyn and *F. meleagris* L. [37,38,40]. However, our findings are consistent with reports of self-infertility as in other closely related species, including *F. aurea* Schott and *F. camtschatcensis* L. [1,46]. In this regard, it would be interesting to explore how incompatibility systems evolve in these species and perennials. On the other hand, the fruit set was not detected in spontaneous self-pollination, induced self-pollination, and apomixis applications. The data of three pollination treatments, in which fruit-set did not occur, indicated that *F. michailovskyi* had limitations for pollination and the reproductive biology was only obligate outcrossing. It was also determined that not only hercogamy but also dichogamy events did not occur in the life cycle of flowers of this species. Self-pollination did not occur in the flower, despite factors such as a transient period of overlap between male and female fertility, as well as the positioning of both the stamen and the pistil in the flower as opposed to hercogamy. The predominant self-infertility of *F. michailovskyi* is likely due to a preference for both protection and maintaining its genetic inheritance and variability, which is consistent with the results that the degree of xenogamy can vary mostly within a single plant depending on various factors, namely pollinator abundance, dichogamy degree, and pollen viability time [47]. It may be stated that insect pollinators have a key role in promoting cross-pollination in *F. michailovskyi* plants. These data reveal that it can be considered a plant species that is functionally heterostylous or characterized by a self-infertility system. This assumption is in line with the findings that avoidance of inbreeding is necessary to ensure efficient pollen exchange in evolutionarily inappropriate mating [48,49].

Based on our natural population pollination experiments, P/O and SII, we can state that *F. michailovskyi* contained high levels of self-incompatibility in reproductive biology. This result indicated that wind or vector insects must be absolute for the species to pollinate flowers, which is consistent with the results reported in some of the *Fritillaria* species [1]. Besides, given the hypothesis that the perianth is structurally reverse growing and that pollen could not reach the stigma either by selfing or by wind due to this feature, we assume that *F. michailovskyi* may not self-pollinate. It has, indeed, been stated that the whole Liliaceae family, like *Fritillaria*, is mostly cross-pollination in general [19]. *F. michailovskyi* has, however, remarkably self-infertile, which does not guarantee sexual reproduction in the absence of insect vectors. These findings were not usually in agreement with the previous report on self-pollinating fritillary species, namely *F. cirrhosa* D.Don [50], *F. persica* L. [51], *F. meleagris* [37], *F. maximowiczii* [38], *F. delavayi* Franch. [40] and *F. koidzumiana* Ohwi [52]. No specific studies have, however, been conducted in this study to determine which factors are more important in the definition of self-incompatibility in *F. michailovskyi*. In addition to those discussed above, expanding our knowledge of the seed viability dynamics and germination fraction of *F. maximowiczii* allows us to not only better model effective population sizes but also describe their demographic dynamics. Considering the TTC test findings, seed viability under spontaneous cross-pollination, xenogamous pollination, and open pollination were 64%, 74%, and 63%, respectively (Table 2). This result, coupled with findings of the ex-situ treatments, demonstrating that the species can produce fully developed seeds by self-pollination, overlaps the opinion on the facultative xenogamy nature of the *F. meleagris* reproduction system, as reported, for example, in Hedstrom [53]. Indeed, seed sets after xenogamous pollination (by hand) and spontaneous cross-pollination are partially small (9 and 8% compared to a seed set of free-pollinated flowers). This is probably due to the fact that pollination is more guaranteed with wind and insect support (Table 2).

*F. michailovskyi* is a remarkable ornamental trait in terms of its flower nodding, bell-shaped and flashy color. *F. persica*, *F. imperialis* L., and *F. aurea* L. species have the most attractive flowers and showiest among the *Fritillaria* species in Turkey, followed by *F. michailovskyi*. Both the attractive and showy flower features of this species caught our attention, and a new cultivar, "Aslay" (registration number F36025/a1) from *F. michailovskyi,* was developed after about 15 years of breeding studies [26–28]. In addition, the perianth segments of the *F. michailovskyi* are yellow at the apex and brownish purple at the base. And thus, this feature allows *F. michailovskyi* to be easily distinguished from other *Fritillaria* species during the fruiting phase. Regarding our flowering dynamic results, the blossoms of the *F. michailovskyi* remained open from about March 16–20 to April 6–15 each year (Figure 1), and pollen presentation lasted for 5.0–6.0 days, which was similar to the *Fritillaria* species reported in previous works [1,37,38]. Our results show that, unlike the perianth shape and perianth color of many *Fritillaria* species in the literature [37–39,50], the flowers have a partially wide bell appearance when opened, and the perianth color is distinctly different at the tip and stem (Figures 1B and 2a). The flowering process took 14–23 days, and this difference was probably owing to temperature fluctuations in the area of plants in March and April. Partial similarities and slight asynchrony were observed between the last studied *Fritillaria* species from blooming to fruiting, which can be explained by the fact that they are due to micro-habitat and climatic differences as reported in previous studies [1,38,40]. On the other hand, the prerequisite for successful pollination and seed formation in flowering plants is pollen viability and stigma receptivity. These two reproductive traits play a key role in understanding the reproductive success of the plant species and in the successful implementation of breeding studies [54,55]. Based on our findings, stigma receptivity of *F. michailovskyi* was highest at the full dehiscence phase (Figure 1E), lowest at the fresh perianth stage (Figure 1G), and decreased gradually throughout the perianth wilting period (Figure 1H). The stigma receptivity of *F. michailovskyi* first increased. Then decreased with the flowering stage, which is in line with other reports [39,50]. It had the strongest stigma receptivity, and its duration was 5–7 days (Figure 3), which was consistent with previously

reported results when compared to other plants of the genus *Fritillaria* [54,55]. On the other hand, the pollen viability of *F. michailovskyi* rose at first and then reduced with the progression of the life cycle. Pollen grains showed high viability in the pre-dehiscence stage (Figure 1C), first dehiscence stage (Figure 1D), and full dehiscence stage (Figure 1E), and then decreased to 55% during the perianth wilting period (Figures 1H and 3). It is an expected feature that the pollen viability rate differs from the results obtained for other *Fritillaria* species and is likely to include characteristics specific to this species. It has, indeed, been reported that pollen production in a flower depends on some factors, such as pollen grain size, anther length, season, and anther separation mode [56]. Recent research has pointed out that the effects of pollen quality and quantity on pollination and fertilization success play a key role in plant protection [57–60], as they are tightly related to genetic and demographic factors that affect population extinction and decline.

Regarding pollinator results of this species, the main pollinator is *A. mellifera*, followed by *B. terrestris* (Figure 4). Compared to other bee species, they are known as more effective pollinators since they collect pollen from various pollen sources and are more generalist resulting create a heterospecific pollen load on the stigmatic surface [61]. Pollinators visited the flowers at different times during the day after the flowers opened. They played an important role as the main actors of spontaneous cross-pollination. It required less effort for pollinators to penetrate the petals of *F. michailovskyi*, as the nectar was located in a larger space inside the bell-shaped corolla and at the base of the tepals. Moths, which are rarely observed during the day, are considered to be nectar thieves, unlike the two pollinators mentioned; however, they were not considered pollinators. These pollinators can feed on nectar, possibly by biting outside of the perianth base where the nectar is located, confirming the pollination deficit and higher yield of this species. We hypothesize that bumblebees are preferred to flowers of *F. michailovskyi* because of their nectar content of it, both having an overall balance between fructose and glucose and nectars of normal size. This hypothesis is in concordance with the result of Knuth [62], Rix and Rast [63], and Yıldız et al. [1], who observed visits of *A. mellifera* and *B. terrestris* in *Fritillaria* species. Based on our results, a difference was also detected in both the duration and frequency of flower visitors, as insects had no trouble foraging on their pendulous flowers (Figure 4). It has been reported that most of the *Andrena* and *Lasioglossum* spend more time in flowers when compared to bumblebees, which supports our results [37]. In addition, honey bees (*A. mellifera*) within the flowers of *F. michailovskyi* tended to move more chaotically. They sometimes completely covered the stigma with pollen, and thus similar stigmas were loaded with more pollen. Partial elucidation of the processes underlying the dependent shifts between pollination behaviors of *F. michailovskyi* agrees with these findings. These findings were similar to *F. aurea* pollinators [1], whereas our results were also different from the results that *Fritillaria* species, namely *F. meleagris*, *F. maximowiczii*, and *F. delavayi* were pollinated by spiders, bumblebees, and bumblebee queens [37,38,40]. In the study area, since the number of other pollinators is less than the honeybee population, we hypothesize that honeybees visit *F. michailovskyi* flowers more than other bees for this reason, although we cannot explain this assumption with our current knowledge.

## 5. Conclusions

This study represents the first paper on the reproductive strategies of *F. michailovskyi*. The current research has also examined a detailed description of various aspects of the reproductive strategy of *F. michailovskyi*, an endemic species, such as P/O, SII, breeding systems, flower biology, phenology, and flower visitors. The flowers of *F. michailovskyi* were generally nodding and bell-shaped, and thus, anthers and nectarines were not visible to approaching pollinators. Our findings indicated the presence of some flower visitors, such as *A. mellifera* and *B. terrestris*, as well as high levels of xenogamy and self-incompatibility of *F. michailovskyi*. The P/O ratio was found to be 0.3, while the SII was set at 0.8, indicating a moderate level of self-incompatibility. Considering that the evolutionary success and survival of *F. michailovskyi* are mostly detected by the efficiency of its reproductive perfor-

mances, this detailed study results of reproductive biology are extremely important. Here, precious information presented on the reproductive strategy of the *F. michailovskyi* endemic species not only has key implications for the management and conservation of threatened *F. michailovskyi* but also may be useful for pollination ecology and breeding system studies in numerous endangered *Fritillaria* species. Finally, it should not be overlooked that pollinators play a key role in the pollination strategy in terms of reproductive biology while breeding studies due to the fact that this species is self-incompatible. Finally, since this species is self-incompatible, it should not be overlooked that pollinators have a key role in the pollination strategy when breeding studies are carried out.

**Supplementary Materials:** The following supporting information can be downloaded at: https://www.mdpi.com/article/10.3390/d15030414/s1, Video S1: video re-cording based on insect activity.

**Author Contributions:** M.A., O.K. and F.Y. designed the experiments and analyzed the data. O.K. and C.B.-N. wrote the manuscript. F.Y., O.K., M.A. and C.B.-N. reviewed the manuscript. All authors have read and agreed to the published version of the manuscript.

**Funding:** The paper was supported both by the General Directorate of Agricultural Research and Policies, Scientific Research Project-SRP (Project No. TAGEM/bbad/17/a09/p09/05) appropriated to Erzincan Horticultural Research Institute, Turkey, and by TUBITAK (The Scientific and Technological Research Council of Turkey), project No. 106G022 and 110G007.

**Institutional Review Board Statement:** Not applicable.

**Informed Consent Statement:** Not applicable.

**Data Availability Statement:** Data may be found in the paper and Supplementary Materials.

**Acknowledgments:** The Erzincan Horticultural Research Institute and the TUBITAK (The Scientific and Technological Research Council of Turkey) are gratefully acknowledged for their support during our fieldwork. All individuals included in this paper have consented to the acknowledgement.

**Conflicts of Interest:** The authors do not have any conflict of interest.

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
