# Peer review of "Reproductive Biology and Pollination Ecology of Fritillaria michailovskyi Fomin (Liliaceae), Endemic to East Anatolia (Turkey)"

_diversity, doi:10.3390/d15030414_

Round 1
Reviewer 1 Report
The manuscript by Aslay et al describes the flowering and pollination biology of an endemic plant species (Fritillaria michailovskyi).
Overall, I consider the manuscript adequate. A few minor things (mentioned below) need to be corrected. I only feel that it is biggest mistake the extremely long lists of various characters. These should be eliminated because they are very confusing. These should be combined under a few extended terms and used only as such in the next, for example: Flower characteristics (flowering period, inflorescence, flower, length of tepal, length of flament, flower colour flower length), Odour, Nectar, Reproductive organs parameters (average number of ovules/ovary, the mean number of pollen grains per flower, length of anther, length of stamen, length of style, pollen-ovule ratio, pollen surface, pollen shape, pollen cell, pollen size, pollen viability, stigma surface).
Minor things:
L18, 105: „Turkey” instead of „Turkiye”
L44, 47, 112, 148: „phenological” instead of „phonological”. The „phonological” means someting else: Phonological definition, of or relating to phonology, the study of the distribution and patterning of speech sounds in languages generally.
L170: use symbol × instead of the letter 'x' („10x”)
L219: a space is missing „16-33 mm” instead of „16-33mm”
L220: like above, a space is missing „5-11mm”
L223-227: I understand this sentence, but it can be confusing.
L281: highlight in bold the "F)."
L284-286: please, the types of marking letters on the figures (Fig 1. and Fig 2.) should be uniform.
L298, 384, 415: Double space.
L493: only the scientific name must be written in Italics „to F. aurea”
L497: The scientific name must be written in Italics. „F. michailovskyi”
The sub-figures should be labeled uniformly in the case of Fig 1 and Fig 2. And these markings should be simple in a traditional style (for example: a black letter in a white square placed in a corner of the image).
Author Response
Dear Reviewer,
Thank you very much for your concern and support for our manuscript entitled “Reproductive biology and pollination ecology of Fritillaria michailovskyi Fomin (Liliaceae), endemic to East Anatolia (Turkey) ”(diversity-2251095). You have provided so many valuable comments, which are very important for the improvement of our manuscript and the following research. We have read your suggestions carefully, and have revised and responded to the manuscript one by one. The same wording and format questions have only been answered once, but the full text has been revised. Please download the revised manuscript and review it again.
Best regards!
Reviewer 2 Report
After analyzing the manuscript entitled:Reproductive biology and pollination ecology of Fritillaria michailovskyi Fomin (Liliaceae), endemic to East Anatolia (Turkey), by Meral Aslay, Faruk Yıldız, Claudia Bita-Nicolae , Ozkan Kaya I make the following considerations:
Abstract-The abstract is direct and very well-written. However, I believe the authors could make it clearer about the findings that are still unprecedent in literature.
Keywords-Replace words that are already in the title.
Introduction-The introduction has a very clear and assertive division of the topics approached. It firstly describes essentials concepts for the comprehension of this work and then the authors describe the current extinction risk and importance of the flora studied. The objects and their importance are also very clear. I believe no changes are needed.
Materials and Methods- The methodology needs some minor changes. The authors could consider making an image to summarize all the experiments, once many were conducted.
Plant material and study site
What is the geographic coordinate of the collection site of plant material?
Line 108-111. I believe this part should be in the discussion as a information for extra results comparison.
Detection of pollination strategies
Line 131-132 - why did you conduct this experiment using 10 plants less?
Line 136-how many days?
Detection of flowering dynamics
Line 142-145-Here, the authors are actually describing their results. It is necessary to replace this part for an clear explanation of how the observations were made and what were the criteria used for the for the classification of the flowering in 8 stages. Did the authors used any literature reference?
Detection of pollen viability and stigma receptivity
line 148-150- describe each of the phases, that is, what you considered as the final dehiscence stage, full dehiscence, first dehiscence, stage, pre-dehiscence stage, and stage initial flowering. One more question, were these stages evaluated in days within the phenological stage?
Line 160-162-It would be interesting to build a table within these classifications;
Detection of floral visitors
Line 198-200-This is quite confuse, if the authors observe the visitors for 20 conscutive days it happened during one specific year, what year and months then?
I suggest videos of insect activity as supplementary material.; What type of trap is used to capture floral visitors?
Results
Please, try to avoid big paragraphs once it difficult the reading and understanding of information. Break all the big blocks in small paragraphs of specific information;
Line 226. Insert the unit of measure in the data when talking about length (mm or cm?);
Line 228. I think you meant 9:30 a.m - 13:30 p.m.;
I didn't like the distribution of figures 2 a, b, c..... which ends up being very
disorganized. My suggestion would be: figure a in place of f and the others
positioned below figure a in sequential order, b, c, d.....
Line 305. Check if it is figure 2f, because in the image figure 2f corresponds to the grains of polen, I believe you meant figure 2h. Check the writing of the text with the image;
Transfer figure 1 after line 251; Observe the positioning of the other figures and tables;
Line 314 and 315: verify. I think you meant... In our results, we observed differences in seed germination between spontaneous cross-pollination, open pollination, xenogamous pollination and with rates of 45, 43 and 51%, respectively;
Check and correct the text. Line 319 and 320: Correct is 9:30 a.m. and 12:00 p.m. ; 13:00 p.m. and 3:00 p.m. ; Check the writing in the text and also in figure 4.
Discussion
The discussion interestingly relates the morphological and physiological findings to the practical part of the plant's ecological relationships, it also highlights the unprecedented findings. However, the discussion is too large and the reading is difficult and tiring. The authors might consider making cuts in repetitive parts; summarizing information of the parameters evaluated in tables to then discuss the ecological relationships; and break the paragraphs into several minors, making it easier to read;
Figure 4. Why is this figure positioned here? This is not clear. The title must be edited and the period of observation is different from the methodology;
Line 460. You didn't mean figure 1E?
The conclusion is well written, however, I suggest that it is interesting to add other important work data, such as results and observed values . And finish with the clarification of the importance of this original study for biology in the plant species, based on its results.
Check the standardization of references. Mainly the names of the journals according with Diversity standards.
Author Response
Dear Reviewer
Our deepest gratitude goes to the anonymous reviewer for their careful work and thoughtful suggestions that have helped improve the manuscript entitled “Reproductive biology and pollination ecology of Fritillaria michailovskyi Fomin (Liliaceae), endemic to East Anatolia (Turkey)” All suggestions were carefully reviewed and changes were made to the attached file. Thank you again for your comments and we look forward to hearing from you regarding our submission. We would be glad to respond to any further questions and comments that you may have.
Best regards!